# Two-Axis Optoelectronic Stabilized Platform Based on Active Disturbance Rejection Controller with LuGre Friction Model

Xueyan Hu, Shunjie Han *, Yangyang Liu and Heran Wang

School of Electrical and Electronic Engineering, Changchun University of Technology, Changchun 130012, China
* Correspondence: hanshunjie@ccut.edu.cn

**Abstract:** To realize the stable tracking control of the optoelectronic stabilized platform system under nonlinear friction and external disturbance, an active disturbance rejection controller (ADRC) with friction compensation is proposed to improve the target tracking ability and anti-disturbance performance. First, a nonlinear LuGre observer is designed to estimate friction behavior and preliminarily suppress the interference of friction torque on the system. Then, an ADRC is introduced to further suppress the residual disturbance after friction compensation, and the stability of the ADRC system is also proved. The effectiveness of this scheme is proved by simulation experiments, and this scheme is compared with conventional ADRC and LuGre friction feedforward compensation. The simulation results show that an ADRC with LuGre friction compensation is better with trajectory tracking performance, which suppresses the influence of disturbance and improves the stability of the optoelectronic stabilized platform system.

**Keywords:** optoelectronic stabilized platform; active disturbance rejection controller; LuGre friction model; parameters identification; disturbance compensation

## 1. Introduction

In recent years, optoelectronic stabilized platforms have been equipped on various aircrafts, ships and other carriers and are widely used in military reconnaissance and civil fields. With the continuous development of science and technology, the optoelectronic stabilized platform is applied in aerial, terrestrial and coastal areas to achieve monitoring and tracking of target objects as well as to obtain high-resolution images, so it has strict requirements on the servo control, image processing and other technologies associated with the platform [1,2]. Disturbances such as nonlinear friction, frame coupling, uncertainty of the system model, carrier vibration and wind resistance moment can lead to instability of the platform's visual axis, affecting the imaging quality of the optoelectronic detection equipment and even leading to the loss of the tracked target object [3]. In addition, nonlinear friction causes the platform system to crawl at a low speed resulting in unstable system motion and generating tracking errors when the system is running at high speed [4]. Therefore, control strategies are used to suppress the effects of nonlinear friction and external disturbances on the optoelectronic stabilized platform in order to improve the stability accuracy of the visual axis [5,6].

Nonlinear friction is a major disturbance, which is generated when the motion state between the frame and axis system changes, making the visual axis shift and affecting the tracking of the target object by the optoelectronic detection equipment. In order to achieve high precision and fast control of the platform system, a large number of scholars have proposed various friction models and corresponding friction compensation methods [7,8]. Among them, the LuGre friction model proposed by C. Canudas De Wit in 1995 can accurately describe the static and dynamic characteristics of friction, which is a more perfect dynamic friction model for dynamic friction compensation [9]. In [10], an adaptive backstepping controller with nonlinear friction compensation was designed for accurate

tracking control of hydraulic systems. In [11], a synchronization control strategy with adaptive friction compensation was applied in a wheeled planetary rover. In [12], a novel robust adaptive sliding mode controller with friction compensation was proposed to achieve accurate and stable control of the electro-optical targeting system. In [13], a robust adaptive integral backstepping control strategy based on a modified LuGre friction model was proposed for opto-electronic tracking systems. Friction compensation suppresses the effect of nonlinear friction on the system and improves the tracking accuracy of the control system to some extent. While other disturbances exist in the system, the effect of friction compensation will be limited, and the control performance of the system will be degraded.

Along with the wide application of optoelectronic stabilized platforms, the requirements for platform performance are becoming increasingly strict. The platform performance is mainly divided into disturbance suppression and stable tracking control. At present, various control methods have been proposed to improve the performance of the optoelectronic stabilized platform in multiple research areas, such as PID control, neural network control, adaptive control and sliding mode control. In [14,15], a compound control method based on backstepping sliding mode control and adaptive neural network was proposed to realize high-performance control of the inertially stabilized platform (ISP). In [16], a nonlinear backstepping controller was designed in order to improve the tracking performance of the two-axis stabilized platform. In [17], a compound PID control strategy was applied to realize high-precision tracking of the target object. The above control strategies contain complex and difficult design structures as well as highly complex algorithms. In practical engineering applications, it is difficult to establish an accurate mathematical model because of the nonlinear, strong coupling and uncertainty characteristics of the optoelectronic stabilized platform, which limits the effect of the above controllers on the system and makes it difficult to get excellent promotion and application.

The active disturbance rejection control strategy is a nonlinear control theory proposed by Prof. Han [18], which is an active anti-disturbance control method. It directly regards the unknown disturbances inside and outside the system as the total disturbance, expands the total disturbance into a new expansion state, and estimates and compensates it through the input and output of the system, so as to counteract disturbance and achieve the purpose of anti-disturbance. At present, ADRCs have been widely used in many engineering fields [19–24], such as multi-motor servo control, electro-hydraulic position servo control, guidance law design, quadrotor UAV control and robot motion control. ADRCs do not need to acquire an accurate mathematical model of the system and can estimate the uncertainty of internal parameters and various external disturbances of the system online. It has the characteristics of high control accuracy, fast tracking speed and strong anti-disturbance ability. Combining ADRCs with various control strategies has been successfully applied in the control system of optoelectronic stabilized platforms, such as an ADRC combined with a neural network, adaptive control, fuzzy control and backstepping control. In [25], an extended state observer (ESO) based on adaptive disturbance frequency was designed to improve the anti-disturbance ability of photoelectric stabilized platforms. In [26], an ADRC based on genetic algorithm parameter tuning was proposed to improve the anti-interference performance and stability accuracy of the ISP. In [27], a nonlinear ESO based on the fractional-order sliding mode control was applied to improve the tracking performance of electro-optical tracking systems. In [28], a least mean square based ADRC controller was designed to suppress various disturbances of the ISP. In [29], a modified nonlinear active disturbance rejection control strategy was proposed to improve the tracking performance of ISPs. However, considering the various uncertainties and disturbances encountered in the operation of the actual optoelectronic stabilized platform system, the challenge of ADRC still lies in the insufficient anti-disturbance capability and tracking performance. Among the various disturbances to the platform system, friction torque disturbance is one of the primary factors affecting the accuracy of the motion control system. In the above control strategies, friction disturbance is regarded as a part of the total disturbance of ADRCs,

and there is no analysis and compensation for friction disturbance according to actual friction characteristics.

Aiming at the control system of optoelectronic stabilized platforms, an ADRC with friction compensation was designed in this paper. Using a LuGre observer and an ADRC, the influence of nonlinear friction and external disturbance of the platform control system was decreased, and the target object can be tracked accurately. The contributions of this paper are given as follows:

First, the LuGre observer with friction compensation and the ESO were applied in the optoelectronic stabilized platform control system. The advantage of the proposed method is that it can observe and suppress the influence of nonlinear friction and external disturbances on the system.

Second, the active disturbance rejection control strategy was used to improve the tracking accuracy of the system, and the stability of the closed-loop system was proved. The ADRC with LuGre friction compensation used in the optoelectronic stabilized platform control system is the most significant feature as compared with conventional ADRCs and LuGre friction feedforward compensation [26,30].

The rest of this paper is organized as follows: First, Section 2 introduces the system dynamics model. Section 3 describes the LuGre friction model and introduces the parameter identification method of the LuGre model. Then, Section 4 is devoted to developing the active disturbance rejection controller, and the stability of the ADRC system is proved. Section 5 verifies the superiority of this method through a MATLAB simulation experiment. Finally, Section 6 gives some conclusions and a discussion of future work.

## 2. Dynamics Model of Optoelectronic Stabilized Platform Systems

### 2.1. Closed-Loop Control System

The optoelectronic stabilized platform in this paper adopted a two-axis four-frame structure, as shown in Figure 1. From outside to inside, they were—in order—the outer azimuth gimbal A, the outer pitch gimbal E, the inner azimuth gimbal a and the inner pitch gimbal e. The photoelectric detectors and photoelectric sensors were mounted on the inner frame, and the outer frame was the carrier of the inner frame and follows the inner frame, allowing the inner frame to make small angle movements in the azimuth and pitch axis. The main function of the outer frame of the platform was to carry the inner frame and the photoelectric equipment, and to provide a completely enclosed environment for the inner frame. When the optoelectronic stabilized platform is subjected to huge external disturbances, these disturbances must go through the isolation of the outer frame to transmit to the inner frame, which greatly weakens the disturbances transmitted to the inner frame. The inner frame, as the carrier of the photoelectric equipment, mainly determines the stability accuracy of the optoelectronic stabilized platform.

In order to improve the anti-disturbance performance and tracking accuracy of the platform, the platform control system adopted a three closed-loop control scheme consisting of current loop, stabilization loop and position loop. The single-gimbal closed-loop control system is shown in Figure 2. Among them, the current loop was used to reduce the influence of current fluctuation on motor torque output. The stabilization loop was the core of the control system, which was the key to stabilize the visual axis of the platform and improve the anti-disturbance performance of the system. The position loop compensated the distance between the controlled target and the visual axis by taking the miss distance as the control quantity, realizing accurate tracking.

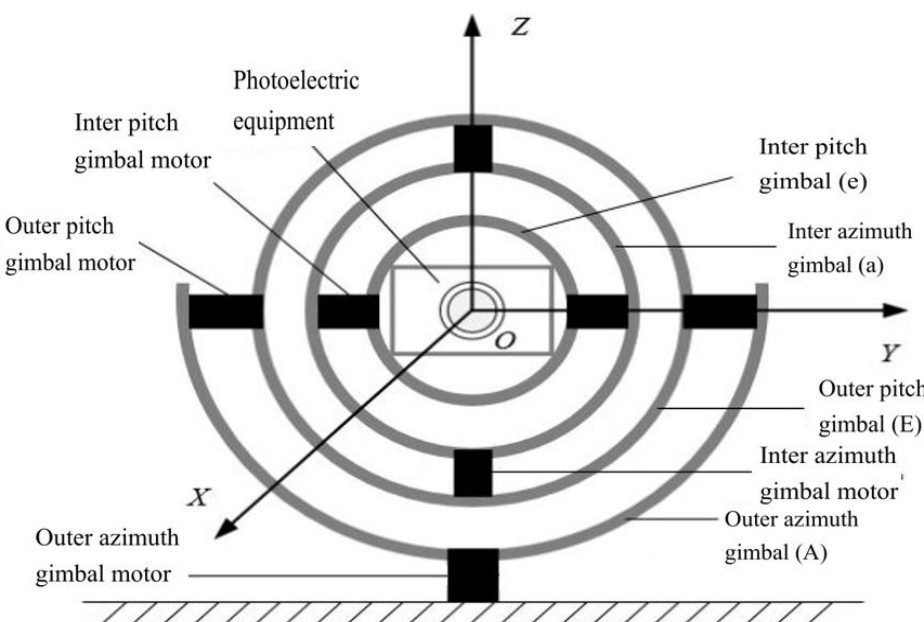

**Figure 1.** Schematic diagram of two-axis four-frame optoelectronic stabilized platform.

The control system consists of current sensor, rate gyro, coder, torque motor, pulse-width modulation (PWM) and controllers. The blocks of $G-pos$, $G-spe$ and $G-cur$ stand for the controllers in the position loop, stabilization loop and current loop, respectively. The PWM block was used to amplify power to drive the torque motor. Rate gyro and coder were used to measure the angle speed and angle of the frame when it moved. $\theta_{in}$ is the input angle, and $\omega_{out}$ and $\theta_{out}$ are the output angle speed and angle of the system, respectively; $M_f$ is the friction torque, and $M_d$ is other disturbances except friction. The LuGre model is introduced in the next section.

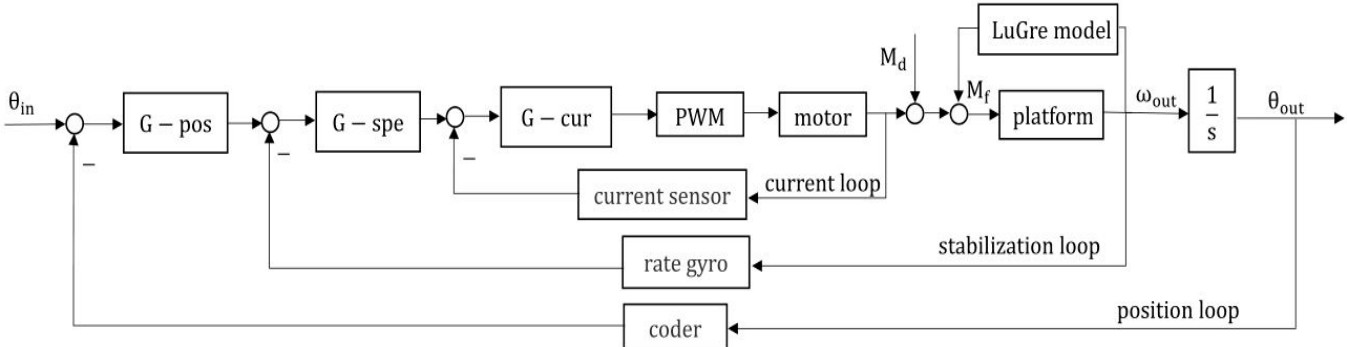

**Figure 2.** Block diagram of three closed-loop control system for single-gimbal.

### 2.2. Motor System

The four frames of the optoelectronic stabilized platform are driven independently by four torque motors. Assuming that the load frame is rigid, according to the motor voltage balance equation and torque balance equation, the mathematical model of the motor system is obtained as follows:

$$
\begin{cases}
\dot{\theta} = \omega \\
U_a = C_e\omega + I_aR_a + L_a\dot{I}_a \\
M_m = C_mI_a \\
J\dot{\omega} + D_l\omega = M_m - M_f - M_d
\end{cases}
\tag{1}
$$

where $\theta$ and $\omega$ are angle and angular velocity, respectively; $U_a$, $I_a$, $R_a$ and $L_a$ are the armature voltage, current, resistance and inductance of the motor, respectively; $C_e$ represents

the back-EMF coefficient; $M_m$ is the electromagnetic torque of the motor; $C_m$ represents the torque coefficient of motor; $J$ is the total moment of inertia of the motor system; and $D_l$ is the damping coefficient of frame rotation.

In general, the value of $D_l$ is so small that it can be neglected. In the actual system, the current loop bandwidth is much larger than the speed loop bandwidth, and the back-EMF is a slowly changing disturbance to the current loop, so the effect of the back-EMF can be disregarded when designing the current loop. Figure 3 shows the structure diagram of the current loop; the current controller often uses a PI controller with a transfer function of $K_i \frac{\tau_i s + 1}{s}$, where $K_i$ is the gain of the current controller, and $\tau_i$ is the integration time constant. The transfer function of the PWM is $\frac{K_p}{T_p s + 1}$, where $K_p$ is the power amplification coefficient and $T_p$ is the switching period of the PWM square wave. $K_f$ is the current feedback coefficient.

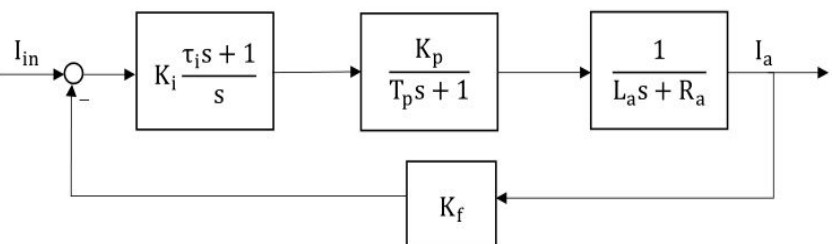

**Figure 3.** Current loop control block diagram.

Taking $\tau_i = T_e = L/R$, the inertia link corresponding to the electromagnetic time constant in the motor can be cancelled. Let $\frac{R}{K_i K_p K_f} = 2T_p$ when the damping ratio is $\xi = 0.707$; the system is second-order optimal. Since the conduction time constant is usually small, the current closed-loop transfer function can be simplified as:

$$G_i = \frac{1/K_f}{Rs\left(T_p s + 1\right)/K_i K_p K_f + 1} = \frac{1/K_f}{2T_p^2 s^2 + 2T_p s + 1} \approx \frac{1}{K_f\left(2T_p s + 1\right)} \tag{2}$$

The LuGre friction model is used to compensate the system considering the friction torque to which it is subjected.

## 3. LuGre Model Parameter Identification

### 3.1. LuGre Model

Based on [9,31,32], the LuGre friction model can be given by:

$$\begin{cases} M_f = \sigma_0 z + \sigma_1 \dot{z} + \sigma_2 \omega \\ \dot{z} = \omega - \frac{|\omega|}{g(\omega)} z \\ \sigma_0 g(\omega) = M_c + (M_s - M_c)e^{-\left(\frac{\omega}{\omega_s}\right)^2} \end{cases} \tag{3}$$

where $z$ is the average deflection of bristles; $\sigma_0$ is the stiffness coefficient of bristles; $\sigma_1$ is the damping coefficient; $\sigma_2$ is the viscous friction coefficient; function $g(\omega)$ is positive and represents different friction effects such as Stribeck effect; $M_c$ represents coulomb friction torque; $M_s$ represents static friction torque; and $\omega_s$ represents the Stribeck angular velocity.

The parameter identification of the LuGre model contains static parameter identification and dynamic parameter identification [33,34], which respectively describe the friction behavior of the system in steady state and critical state, and static parameters are estimated by the Stribeck curve.

### 3.2. Static Parameter Identification

When the system is in steady state ($\dot{z} = 0$), the steady-state relationship between total friction torque $M_{sf}$ and angle speed is shown in Equation (4). The system motion velocity is kept constant, for the total friction torque $M_{sf}$ is equal to the control torque $M_m$.

$$M_{sf} = \left[M_c + (M_s - M_c)e^{-\left(\frac{\omega}{\omega_s}\right)^2}\right] sgn(\omega) + \sigma_2\omega = M_m \tag{4}$$

Given that the platform system rotates at a set of invariable angle speeds $\{\omega_i\}_{i=1}^N$, and the corresponding control torque is $\{M_{mi}\}_{i=1}^N$, the steady-state correspondence between the friction torque and angle speed, i.e., the Stribeck curve, is determined. The parameter vectors to be identified are as follows:

$$P_1 = \left[\hat{M}_s, \hat{M}_c, \hat{\omega}_s, \hat{\sigma}_2\right] \tag{5}$$

The identification error is defined as:

$$e(P_1, \omega_i) = M_{mi} - M_{sf}(P_1, \omega_i) \tag{6}$$

where $M_{sf}(P_1, \omega_i)$ denotes the desired friction torque, which is determined by Equation (7).

$$M_{sf}(P_1, \omega_i) = \left[\hat{M}_c + (\hat{M}_s - \hat{M}_c)e^{-\left(\frac{\omega_i}{\hat{\omega}_s}\right)^2}\right] sgn(\omega_i) + \hat{\sigma}_2\omega_i \tag{7}$$

The objective function is defined as

$$S_1 = \frac{1}{2}\sum_{i=1}^N e^2(P_1, \omega_i) \tag{8}$$

Ultimately, the static parameters $M_c$, $M_s$, $\omega_s$ and $\sigma_2$ are determined by minimizing the objective function $S_1$.

### 3.3. Dynamic Parameter Identification

The state variable $z$ of the LuGre friction model cannot be measured directly, so the values of $\sigma_{00}$ and $\sigma_{10}$ are approximated by Equation (9) using the presliding displacement method [35], and then the dynamic parameters $\sigma_0$ and $\sigma_1$ are obtained by the least squares method.

$$\begin{cases} \sigma_{00} \approx \frac{\Delta M}{\Delta\theta} \\ \sigma_{10} \approx 2\sqrt{\sigma_{00}J} - \sigma_2 \end{cases} \tag{9}$$

where $\Delta M$ and $\Delta\theta$ are the variation of control torque and angle in the presliding phase, respectively. The parameter vectors to be identified are as follows:

$$P_2 = [\hat{\sigma}_0, \hat{\sigma}_1] \tag{10}$$

The identification error is defined as

$$e(P_2, t_i) = \theta(t_i) - \theta(P_2, t_i) \tag{11}$$

where $\theta(t_i)$ represents the angle of the system output corresponding to moment $t_i$ and $\theta(P_2, t_i)$ represents the angle of the model system consisting of the identification parameters corresponding to moment $t_i$.

The objective function is defined as:

$$S_2 = c_1\sum_{i=1}^N e^2(P_2, t_i) + c_2 max\{|e(P_2, t_i)|\} \tag{12}$$

where $c_1$ and $c_2$ are weight coefficients. The dynamic parameters $\sigma_0$ and $\sigma_1$ are determined by minimizing the objective function $S_2$.

## 4. Controller Design

The LuGre model is put forward to compensate the friction torque disturbance of the platform system. However, because the system will change with the external environment, the system friction characteristics may differ from the friction compensation model. The LuGre model is insensitive to other disturbances, and the compensation result is that there are still residual disturbances in the system. Therefore, the ADRC is introduced to observe and compensate these residual disturbances and accurately track the target objects in real time.

### 4.1. ADRC Algorithm

An ADRC consists of three parts: tracking differentiator (TD), extended state observer (ESO) and nonlinear state error feedback control law (NLSEF). Figure 4 shows the structure diagram of an ADRC.

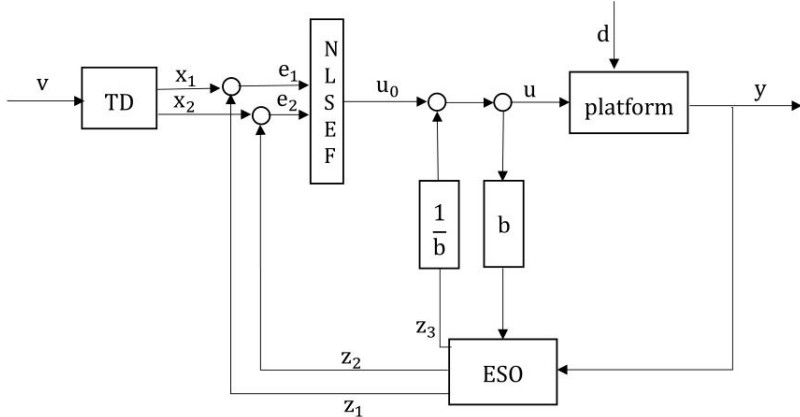

**Figure 4.** Second-order ADRC system.

Where $x_1$ and $x_2$ are the outputs of TD, which respectively represent the tracking signal and differential signal of the input signal $v$; $z_1$, $z_2$ and $z_3$ are the observed values of $x_1$, $x_2$ and $d$, respectively; $u$ and $y$ are the input and output of the controlled object, respectively; and $b$ is the system gain.

According to Figure 2, the disturbance of the platform system is combined with friction compensation, and the disturbance structure diagram of the system after friction compensation can be obtained, as shown in Figure 5. $M'_f$ is the friction torque compensated by LuGre friction, and $\Delta M_f$ is the friction compensation error, with $\Delta M_f = M_f - M'_f$.

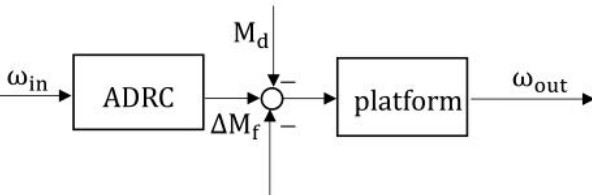

**Figure 5.** Block diagram of system disturbance after friction compensation.

With the current $I$ as the input of the controlled object in ADRC, the total disturbance $d = \Delta M_f + M_d$. Assuming that $d$ is bounded, i.e., $\|d\|_\infty \leq d_m$, and $d_m$ denotes the upper limit value of the disturbance, its second-order nonlinear system equation of state can be written as:

$$\begin{cases} \dot{x}_1 = x_2 \\ \dot{x}_2 = bu + d \\ y = x_1 \end{cases} \tag{13}$$

First, TD is used to arrange a transition process, which can effectively track the input signal and extract a differential signal, improving the conflict between system rapidity and overshoot. For a given speed $\omega_r$ of the system, the TD is designed as:

$$\begin{cases} \dot{x}_1 = x_2 \\ \dot{x}_2 = fhan(x_1 - \omega_r, x_2, r, h) \end{cases} \tag{14}$$

where $r$ is the speed factor which can determine the tracking speed, and $h$ is the filter factor.

$fhan(\cdot)$ is the fast optimal control synthesis function, which has a better arrangement of the transition process of the reference signal so as to keep it from overshooting. The specific form is as follows:

$$\begin{cases} d = rh \\ d_0 = dh \\ y = x_1 - v + hx_2 \\ a_0 = \sqrt{d^2 + 8r|y|} \\ a = \begin{cases} x_2 + \frac{(a_0 - d)}{2} & |y| > d_0 \\ x_2 + \frac{y}{h} & |y| \le d_0 \end{cases} \\ fhan = -\begin{cases} rsgn(a) & |a| > d \\ \frac{ra}{d} & |a| \le d \end{cases} \end{cases} \tag{15}$$

Secondly, ESO is the core of the ADRC, which expands the total disturbance $d$ in the system into a new state variable $x_3$ and observes each of the state variables in the system in the following form:

$$\begin{cases} e = z_1 - \omega \\ \dot{z}_1 = z_2 - \beta_1 e \\ \dot{z}_2 = z_3 - \beta_2 fal\left(e, \frac{1}{2}, \delta_1\right) + bu \\ \dot{z}_3 = -\beta_3 fal\left(e, \frac{1}{4}, \delta_1\right) \end{cases} \tag{16}$$

where $\beta_1$, $\beta_2$ and $\beta_3$ are the observer gains; and $\delta_1$ is the controller adjustable parameter. The bandwidth method is used to determine the observer gains by taking $\beta_1 = 3\omega_0$, $\beta_2 = 3\omega_0^2$ and $\beta_3 = \omega_0^3$, where $\omega_0$ is the observer bandwidth.

$fal(\cdot)$ is a nonlinear function of the following form:

$$fal(x, \alpha, \delta) = \begin{cases} \frac{x}{\delta^{(1-\alpha)}} & |x| \le \delta \\ sign(x)|x|^\alpha & |x| > \delta \end{cases} \tag{17}$$

Finally, NLSEF is used to fit state errors and compensate for perturbations; its form is as follows:

$$\begin{cases} e_1 = x_1 - z_1 \\ e_2 = x_2 - z_2 \\ I_0 = \eta_1 fal\left(e_1, \frac{3}{4}, \delta_2\right) + \eta_2 fal\left(e_2, \frac{5}{4}, \delta_2\right) \\ I_{in} = I_0 - \frac{z_3}{b} \end{cases} \tag{18}$$

where $\eta_1$ and $\eta_2$ are the gains of NLSEF; and $\delta_2$ is the controller adjustable parameter.

The structure of the ADRC with LuGre friction compensation is shown in Figure 6.

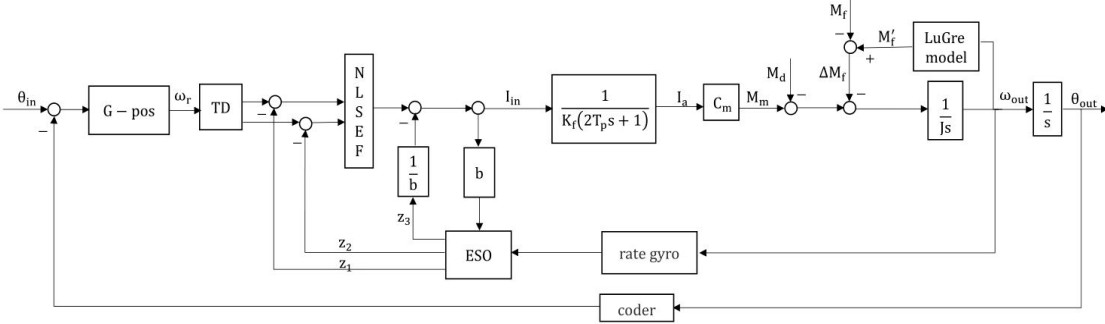

**Figure 6.** Block diagram of the ADRC structure with LuGre friction compensation.

*4.2. System Stability Analysis*

By using nonlinear functions to discuss the ADRC algorithm, the nonlinear ADRC algorithm has many advantages, such as higher accuracy and higher feedback efficiency, but it requires more control parameters to be adjusted, and it is difficult to determine the stability boundary, which is difficult in the theoretical study. Since the core idea of the ADRC is to estimate and compensate disturbance in real time, the linear form is used to study the stability of ADRC systems.

Suppose that the system is given a bounded input signal $r$ and its differential signals $\dot{r}$ and $\ddot{r}$ are bounded, let $\widetilde{e}_i = x_i - z_i, \varepsilon_i = r_i - x_i$. In the linear expansion state, $z_i(i = 1, 2, 3)$ is the estimated value of $x_i$ and $\widetilde{e}_i$ is the estimated error value. The output of ESO $[z_1, z_2, z_3]$ satisfy the following feedback control.

$$u = \frac{k_1}{b}(r - z_1) + \frac{k_2}{b}(\dot{r} - z_2) + \frac{1}{b}(\ddot{r} - z_3) \tag{19}$$

where $k_1$ and $k_2$ are the controller's parameters and are greater than 0. Equation (13) can be described as:

$$\ddot{x} = d - z_3 + k_1(r - z_1) + k_2(\dot{r} - z_2) + \ddot{r} \tag{20}$$

Let $r_1 = r, r_2 = \dot{r}, r_3 = \ddot{r}$ as follows.

$$\begin{cases} \dot{\varepsilon}_1 = \dot{r}_1 - \dot{x}_1 = r_2 - x_2 = \varepsilon_2 \\ \dot{\varepsilon}_2 = \dot{r}_2 - \dot{x}_2 = r_3 - \ddot{x} = -k_1\varepsilon_1 - k_2\varepsilon_2 - k_1\widetilde{e}_1 - k_2\widetilde{e}_2 - \widetilde{e}_3 \end{cases} \tag{21}$$

Let

$$\begin{cases} \varepsilon = \begin{bmatrix} \varepsilon_1 & \varepsilon_2 \end{bmatrix}^T \\ E = \begin{bmatrix} \widetilde{e}_1 & \widetilde{e}_2 & \widetilde{e}_3 \end{bmatrix}^T \\ A_1 = \begin{bmatrix} 0 & 1 \\ -k_1 & -k_2 \end{bmatrix} \\ A_2 = \begin{bmatrix} 0 & 0 & 0 \\ -k_1 & -k_2 & -1 \end{bmatrix} \end{cases} \tag{22}$$

Equation (21) can be expressed in matrix form as follows.

$$\dot{\varepsilon} = A_1\varepsilon + A_2\widetilde{e} \tag{23}$$

The estimation error of ESO is shown as follows.

$$\lim_{t \to \infty} \|\widetilde{e}\|_2 = 0 \tag{24}$$

**Theorem 1.** *Assuming that d is bounded, then there exists controller parameters $k_1 > 0$ and $k_2 > 0$ to make tracking error $\varepsilon$ tend to zero, such that the closed-loop system is stable* [36].

**Proof of Theorem 1.** According to Equation (23), one has □

$$\varepsilon = exp(A_1 t) \cdot \varepsilon(0) + \int_0^t exp(A_1(t - \tau)) \cdot A_2 \widetilde{e} d\tau \tag{25}$$

Due to

$$|\lambda I_2 - A_1| = \lambda^2 + k_2 \lambda + k_1 \tag{26}$$

Choose the values of $k_1$ and $k_2$ such that $|\lambda I_2 - A_1| = (\lambda + \overline{\lambda}_1)(\lambda + \overline{\lambda}_2)$, where $0 < \overline{\lambda}_1 < \overline{\lambda}_2$. $A_1$ has two different eigenvalues, so that $A_1$ can be diagonalized; that is, there is an invertible matrix $T$ such that $A_1 = \overline{T} diag\{-\overline{\lambda}_1, -\overline{\lambda}_2\} \overline{T}^{-1}$, so:

$$exp(A_1 t) = \overline{T} diag\{exp(-\overline{\lambda}_1 t), exp(-\overline{\lambda}_2 t)\} \overline{T}^{-1} \tag{27}$$

For any positive number $t > 0$,

$$exp\|(A_1 t)\|_2 \le \|\overline{T}\|_2 \|\overline{T}^{-1}\|_2 exp(-\overline{\lambda}_1 t) = \overline{\beta} exp(-\overline{\lambda}_1 t) \tag{28}$$

When $\overline{\lambda}_1$ and $\overline{\lambda}_2$ are selected, $\overline{\beta}$ is a constant.

$$\lim_{t \to \infty} \|exp(A_1 t)\|_2 = 0 \tag{29}$$

Similarly,

$$\|exp(A_1(t - \tau))\|_2 \le \overline{\beta} exp(-\overline{\lambda}_1(t - \tau)), t \ge \tau \tag{30}$$

Since the ESO estimation error $\lim_{t \to \infty} \|\widetilde{e}\|_2 = 0$, the second norm of $\widetilde{e}$ has an upper bound $\alpha$. For any specified $\eta > 0$, there is a positive number $t_0$, and the second norm of $\widetilde{e}$ is less than $\eta$ when $t > t_0$.

$$
\begin{aligned}
\|\int_0^t exp(A_1(t - \tau)) A_2 \widetilde{e} d\tau\|_2 &= \|\int_0^{t_0} exp(A_1(t - \tau)) A_2 \widetilde{e} d\tau\|_2 + \|\int_{t_0}^t exp(A_1(t - \tau)) A_2 \widetilde{e} d\tau\|_2 \\
&\le \overline{\beta} exp(-\overline{\lambda}_1 t) \cdot \|A_2\|_2 \alpha \int_0^{t_0} exp(\overline{\lambda}_1 \tau) d\tau + \overline{\beta} exp(-\overline{\lambda}_1 t) \cdot \|A_2\|_2 \eta \int_{t_0}^t exp(\overline{\lambda}_1 \tau) d\tau \\
&= exp(-\overline{\lambda}_1 t) \cdot \overline{\beta} \|A_2\|_2 \alpha \int_0^{t_0} exp(\overline{\lambda}_1 \tau) d\tau + \overline{\beta} exp(-\overline{\lambda}_1 t) \|A_2\|_2 \cdot \frac{\eta}{\overline{\lambda}_1} [exp(\overline{\lambda}_1 t) - exp(\overline{\lambda}_1 t_0)] \\
&\le \overline{M}_1 exp(-\overline{\lambda}_1 t) + \frac{\eta}{\overline{\lambda}_1} \overline{\beta} \|A_2\|_2 \\
&= \overline{M}_1 exp(-\overline{\lambda}_1 t) + \overline{M}_2 \eta
\end{aligned}
\tag{31}
$$

where $\overline{M}_1 = \overline{\beta} \|A_2\|_2 \alpha \int_0^{t_0} exp(\overline{\lambda}_1 \tau) d\tau$ and $\overline{M}_2 = \frac{\overline{\beta} \|A_2\|_2}{\overline{\lambda}_1}$ are constants. From $exp(-\overline{\lambda}_1 t) \to 0(t \to \infty)$, Equation (32) can be obtained as below.

$$\lim_{t \to \infty} \|\int_0^t exp(A_1(t - \tau)) A_2 \widetilde{e} d\tau\|_2 = 0 \tag{32}$$

Combining Equations (25), (29) and (32), Equation (33) is described as follows:

$$\lim_{t \to \infty} \|\varepsilon\|_2 = 0 \tag{33}$$

According to Theorem 1, with the system model unknown, the above assumes that the system total disturbance $d$ is bounded; that is, $\|d\|_\infty \le d_m$, and there are controller's parameters $k_1 > 0$ and $k_2 > 0$ making the tracking error of the closed-loop system tend to 0. Thus, for bounded input $r$, the output of the closed-loop system is bounded; that is, the closed-loop system is BIBO stable.

## 5. Simulations and Results

Because the inner frame plays a major role in the stability accuracy of the optoelectronic stabilized platform, this paper takes the inner pitch frame as an example and simulates the inner pitch frame of the platform with MATLAB to verify the effectiveness of the scheme. The parameters of LuGre model and platform system are listed in Table 1.

**Table 1.** System and LuGre model parameters.

| Parameters | Value |
|---|---|
| Total moment of inertia (kg·m$^2$) | J = 1.1 |
| Motor resistance ($\Omega$) | $R_a$ = 4.2 |
| Motor inductance (mH) | $L_a$ = 3.78 |
| Torque coefficient (Nm/A) | $C_m$ = 3.478 |
| Back-EMF coefficient (V·s/deg) | $C_e$ = 0.78 |
| Stiction force (Nm) | $M_s$ = 0.6 |
| Coulomb friction (Nm) | $M_c$ = 0.15 |
| Viscous friction coefficient (Nm·s/deg) | $\sigma_2$ = 0.02 |
| Stribeck angular velocity (deg/s) | $\omega_s$ = 0.05 |
| Stiffness coefficient (Nm/deg) | $\sigma_0$ = 47.3 |
| Damping coefficient (Nm·s/deg) | $\sigma_1$ = 0.73 |

To demonstrate the superiority of the ADRC with a LuGre friction model, a comparison experiment is made with LuGre model feedforward compensation and an ADRC.

### 5.1. Sinusoidal Trajectory Tracking Experiment

The sinusoidal input signal $\theta_{in} = 2sin(0.5\pi t)$ is the tracking curve of the desired output, and an external disturbance $M_d = sin(3\pi t) + 2sin(2\pi t) + sin\left(t + \frac{\pi}{3}\right)$ is added to the system. The simulation results of the three control methods are shown in Figure 7, and Table 2 compares the angle tracking error and angle speed error of the three control methods in terms of both maximum (MAX) value and root mean square (RMX). Figure 7a represents the angle tracking curve, and it can be seen from the figure that the ADRC with the LuGre model can track the given input signal better. Figure 7b represents the angle error curve. The maximum value of the angle error of this scheme is about 0.0628° and the root mean square of the error is about 0.0171°. The root mean square of the error is reduced by 62.54% compared with the ADRC and 93.14% compared with LuGre model feedforward compensation; therefore, the error is significantly reduced.

**Table 2.** Comparison of angle tracking error and angle speed error under sinusoidal input signal.

| Controller | ADRC | LuGre | LuGre + ADRC |
|---|---|---|---|
| Angle error (MAX) | 0.1884° | 0.4418° | 0.0628° |
| Angle error (RMX) | 0.0323° | 0.1764° | 0.0121° |
| Angle speed error (MAX) | 0.4638°/s | 0.9837°/s | 0.0693°/s |
| Angle speed error (RMX) | 0.0832°/s | 0.3987°/s | 0.0176°/s |

Figure 7c,d represents the angle speed and angle speed error curves; the maximum value of the angle speed error of this scheme is about 0.0495°/s and the root mean square of the error is about 0.0156°/s. The root mean square of the error decreased by up to 78.85% compared with ADRC and 95.58% compared with LuGre model feedforward compensation. The results show that the ADRC with LuGre model has better position tracking capability and the best anti-disturbance effect.

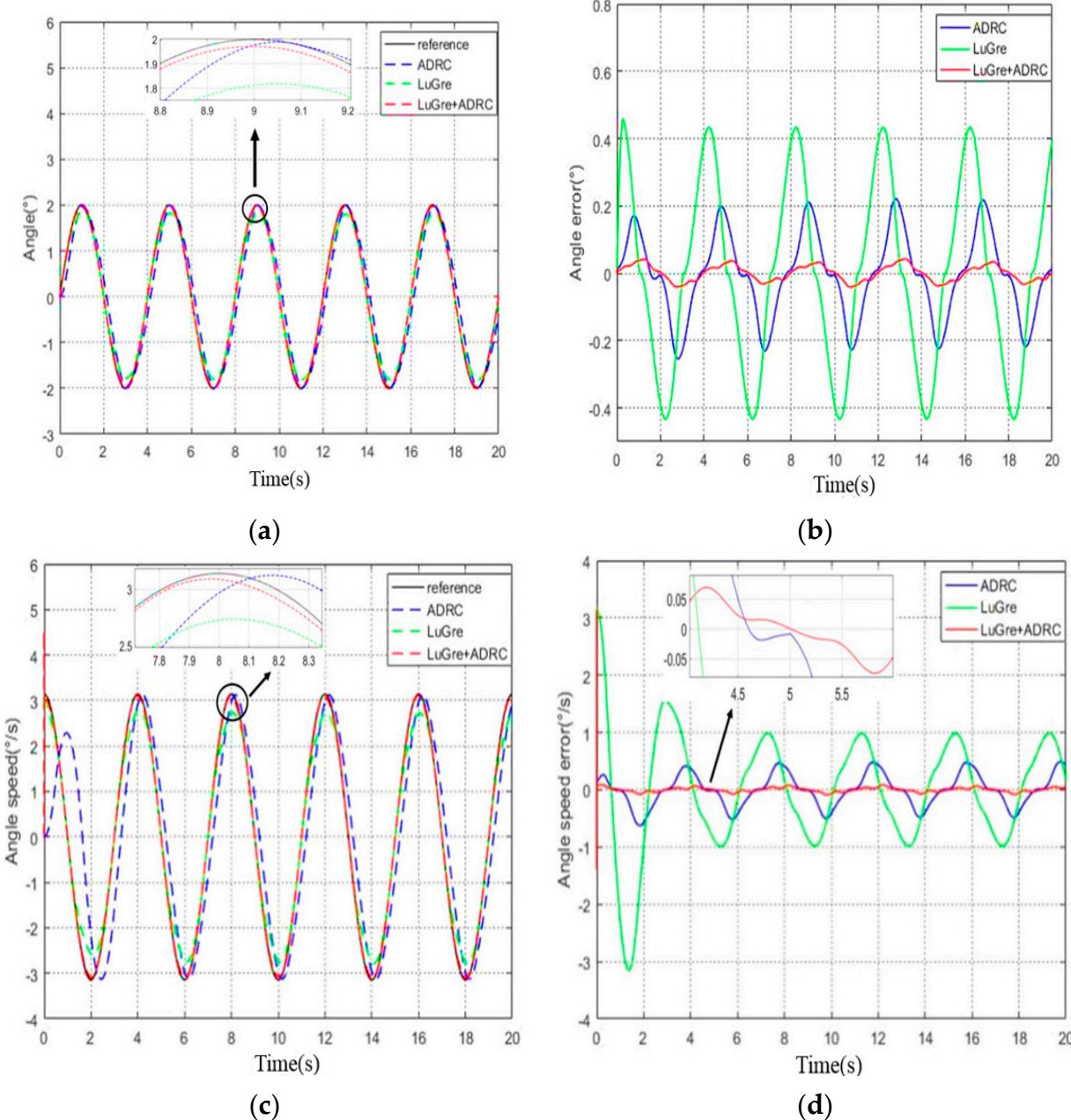

**Figure 7.** Sinusoidal trajectory tracking results. (**a**) Diagram of the angle tracking curve; (**b**) Diagram of the angle error curve; (**c**) Diagram of the angle speed tracking curve; (**d**) Diagram of the angle speed error curve.

### 5.2. Multiple Sinusoidal Trajectory Tracking Experiment

The input signal $\theta_{in} = sin(0.5\pi t) + 2sin\left(t + \frac{\pi}{6}\right)$ is the tracking curve of the desired output, and a random signal with zero mean and unit variance is added to the system as an external disturbance, $M_d = rand(1)$. The simulation results of the three control methods are shown in Figure 8, and Table 3 compares the position tracking error and angular velocity error of the three control methods in terms of both maximum value and root mean square. Figure 8a represents the angle tracking curve, from which it can be seen that the angle $\theta$ of the scheme tracks the given input signal with a small phase difference.

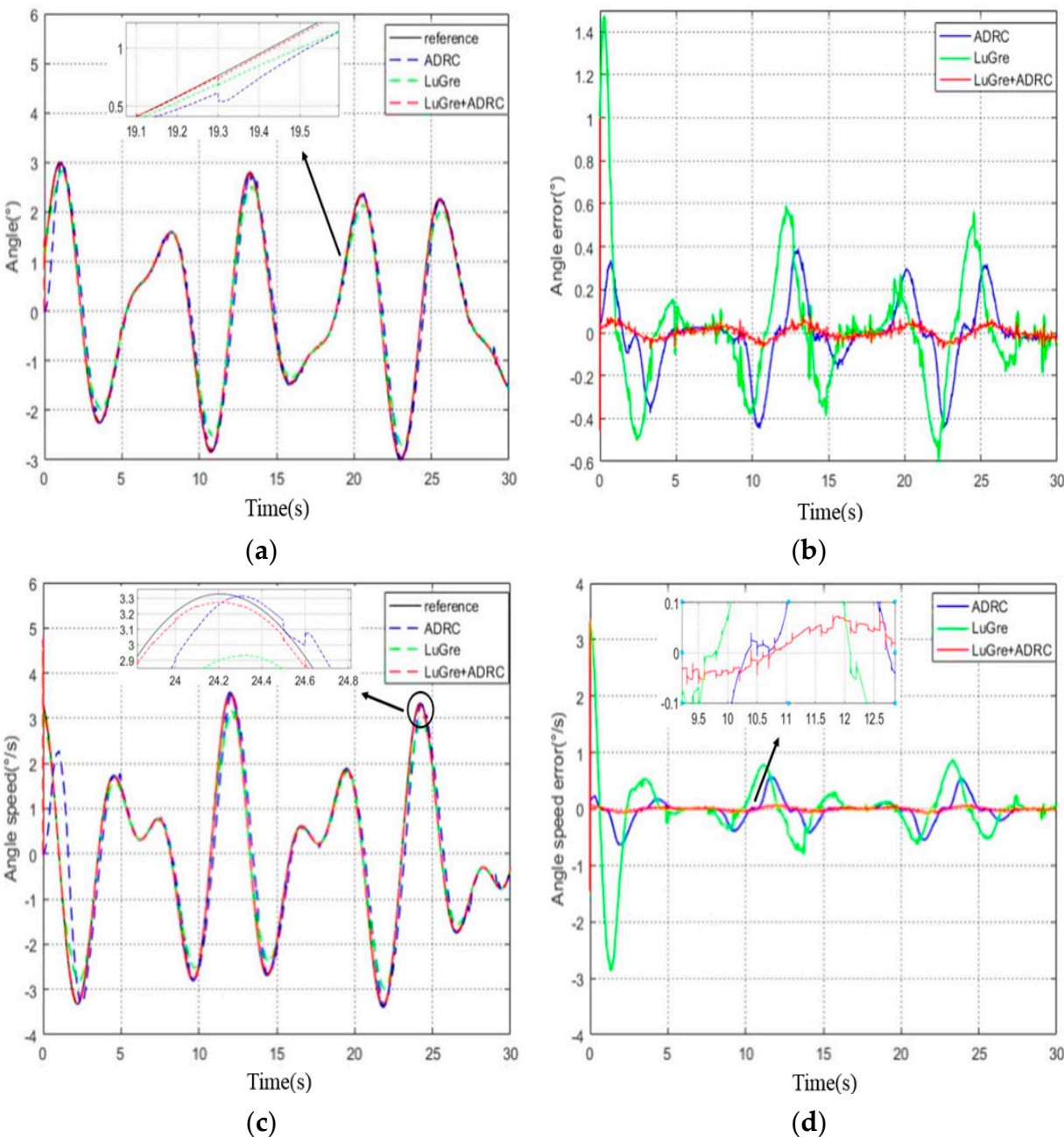

**Figure 8.** Multiple sinusoidal trajectory tracking results. (**a**) Diagram of the angle tracking curve; (**b**) Diagram of the angle error curve; (**c**) Diagram of the angle speed tracking curve; (**d**) Diagram of the angle speed error curve.

**Table 3.** Comparison of angle tracking error and angle speed error under multiple sinusoidal input signals.

| Controller | ADRC | LuGre | LuGre + ADRC |
|---|---|---|---|
| Angle error(MAX) | 0.3801° | 0.5834° | 0.0542° |
| Angle error(RMX) | 0.0752° | 0.1467° | 0.0117° |
| Angle speed error(MAX) | 0.5365°/s | 0.7737°/s | 0.0737°/s |
| Angle speed error(RMX) | 0.0973°/s | 0.1447°/s | 0.0189°/s |

Figure 8b represents the angle error curve; the maximum value of the angle error of this scheme is about 0.0542°, and the root mean square of the error is about 0.0117°; the

root mean square of the error is reduced by 84.44% compared to the ADRC and 92.02% compared to LuGre model feedforward compensation.

Figure 8c,d represents the angle speed and angle speed error curves. The peak value of the angular velocity error of this scheme is about $0.0660°/s$ and the root mean square of the error is about $0.0189°/s$. The root mean square of the error decreased by up to 80.58% compared with the ADRC and 86.94% compared with LuGre model feedforward compensation. The results show that the ADRC with LuGre model has the smallest tracking error and the best anti-disturbance effect.

The above simulation experiments show that the ADRC with LuGre friction compensation proposed in this paper has better anti-disturbance performance and target tracking capability in the case of nonlinear friction and external disturbances, and the control scheme can achieve high precision and stable control compared with conventional ADRCs and LuGre friction feedforward compensation.

## 6. Conclusions

In order to improve the anti-disturbance performance of the optoelectronic stabilized platform, an active disturbance rejection controller with LuGre friction model was proposed to realize stable tracking control in the case of nonlinear friction and external interference. First, the LuGre model was introduced to suppress the disturbance of friction torque on the system. Second, utilizing the characteristics of the ADRC to observe and compensate disturbance, the compensation error of the friction model and the influence of other disturbances on the system were decreased, and the stability of the ADRC system was proved. Finally, under different input signals, the proposed control scheme was compared with the conventional ADRC and LuGre model feedforward compensation. Simulation results show that the ADRC with LuGre friction model proposed in this paper can improve the tracking accuracy and stability of the system. We aim to further improve the ADRC algorithm's control performance and lessen the impact of multi-source disturbances on optoelectronic stabilized platform systems.

**Author Contributions:** Conceptualization, H.W. and Y.L.; methodology, X.H. and S.H.; software, X.H.; validation, X.H., H.W. and Y.L.; formal analysis, X.H.; data curation, S.H.; writing—original draft preparation, X.H.; writing—review and editing, S.H. and X.H.; supervision, S.H.; project administration, S.H.; funding acquisition, S.H. All authors have read and agreed to the published version of the manuscript.

**Funding:** The research was funded by the Jilin Provincial Department of Science and Technology, funding number 20210201025GX. The APC was funded by the Jilin Provincial Department of Science and Technology.

**Institutional Review Board Statement:** Not applicable.

**Informed Consent Statement:** Not applicable.

**Data Availability Statement:** Not applicable.

**Conflicts of Interest:** The authors declare no conflict of interest.

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
