# Peer review of "Two-Axis Optoelectronic Stabilized Platform Based on Active Disturbance Rejection Controller with LuGre Friction Model"

_electronics, doi:10.3390/electronics12051261_

Round 1

Reviewer 1 Report

This paper entitled with ‘Two-Axis Optoelectronic Stabilized Platform Based on Active Disturbance Rejection Controller with LuGre Friction Model’ has proposed a scheme based on active disturbance rejection controller with LuGre friction model to improve the target tracking ability and anti-disturbance performance for two-axis optoelectronic stabilized platform system based on simulation. The results show very big improvement compared with conventional active disturbance rejection controller and LuGre friction feedforward compensation alone. The result is very impressive, and the paper is well-written, well-referenced, and well-organized. I recommend this manuscript to be accepted for publication after addressing the following questions.

1.       What are the drawbacks for the scheme that is proposed in this paper? It seems that the combination of ADRC and LuGre fricition model makes the control system more complex, which may require more extensive tuning and testing to achieve optimal performance.

2.       How is the computational overhead for the proposed scheme? The ADRC relies on an extended state observer to estimate the disturbance, which can be computationally intensive. In addition, the LuGre Friction Model also requires some additional computational overhead. These factors can potentially limit the achievable control bandwidth of the system.

3.       How is the proposed mode uncertainty? The LuGre Friction Model often relies on accurate knowledge of the system's friction characteristics, which may be difficult to obtain in practice. This can introduce some uncertainty into the control system, which may impact its overall performance.

Reviewer 3 Report

The authors of this paper mainly focused on the control structure of the optoelectronic stabilized platform under nonlinear friction and external disturbance. In recent years these platforms are used in several fields for both commercial and non commercial purposes. Though the study carried out is an interesting one, I can not recommend this paper in its present form. My comments are as follows:

1. The last four paragraphs of Introduction are not well connected between lines. The research mainly has objectives and goals, based on which we organize our papers. I urge the authors to choose proper terminology and make connections between lines. 

2. The fonts in all Figures are too small. Bold and italics feature on small fonts makes the figures blurred. Please prepare the figures with more clear fonts.

3. What was the value of the critical gain parameter?

4. What was the value of controller bandwidth?

5. The authors only stated the results. How did the authors claim the accuracy of the results? I can not see any other evidence that can prove the study conducted by the authors is absolutely correct?

6. There are many typographical errors scattered in the ms, specifically while mentioning the Figure number. Please check this error very carefully. It creates difficulty and confusion for readers to understand the claim of the authors.

7. The conclusion part is too short. It must be rewritten with more clarification and future direction.
